

# Defenses of whirligig beetles against native and invasive frogs

Shinji Sugiura[1] and Masakazu Hayashi[2]

[1] Graduate School of Agricultural Science, Kobe University, Kobe, Hyogo, Japan
[2] Hoshizaki Green Foundation, Izumo, Shimane, Japan

## ABSTRACT

Many native insects have evolved defenses against native predators. However, their defenses may not protect them from non-native predators due to a limited shared history. The American bullfrog, *Aquarana catesbeiana* (Anura: Ranidae), which has been intentionally introduced to many countries, is believed to impact native aquatic animals through direct predation. Adults of whirligig beetles (Coleoptera: Gyrinidae), known for swimming and foraging on the water surface of ponds and streams, reportedly possess chemical defenses against aquatic predators, such as fish. Although whirligig beetles potentially encounter both bullfrogs and other frogs in ponds and lakes, the effectiveness of their defenses against frogs has been rarely studied. To assess whether whirligig beetles can defend against native and non-native frogs, we observed the behavioral responses of the native pond frog, *Pelophylax nigromaculatus* (Anura: Ranidae), and the invasive non-native bullfrog, *A. catesbeiana*, to native whirligig beetles, *Gyrinus japonicus* and *Dineutus orientalis*, in Japan. Adults of whirligig beetles were provided to frogs under laboratory conditions. Forty percent of *G. japonicus* and *D. orientalis* were rejected by *P. nigromaculatus*, while all whirligig beetles were easily consumed by *A. catesbeiana*. Chemical and other secondary defenses of *G. japonicus* and *D. orientalis* were effective for some individuals of *P. nigromaculatus* but not for any individuals of *A. catesbeiana*. These results suggest that native whirligig beetles suffer predation by invasive non-native bullfrogs in local ponds and lakes in Japan.

## INTRODUCTION

Invasive non-native predators have been introduced into many countries, significantly impacting native biota through direct consumption (*Goldschmidt, Witte & Wanink, 1993*; *Kenis et al., 2009*; *Sugiura, 2016*; *David et al., 2017*). Many native species are vulnerable to predation by non-native species (*Goldschmidt, Witte & Wanink, 1993*; *Doherty et al., 2016*; *Sugiura, 2016*) because they share a much shorter evolutionary history with these non-native predators (*Fritts & Rodda, 1998*; *Strauss, Lau & Carroll, 2006*; *Carthey & Banks, 2014*). However, some native species can evade or repel non-native predators using pre-existing anti-predator strategies (*Davis, Epp & Gabor, 2012*; *Carthey & Banks, 2014*; *Sugiura & Date, 2022*). Native species have evolved chemical, physical, or morphological defenses against native predators (*Eisner, 2003*; *Eisner, Eisner & Siegler, 2005*; *Sugiura,*

Corresponding author
Shinji Sugiura,
ssugiura@people.kobe-u.ac.jp,
sugiura.shinji@gmail.com

*2020a*), and some of these pre-existing anti-predator strategies can protect native species from non-native predators (*Davis, Epp & Gabor, 2012*; *Sugiura & Date, 2022*). However, only a few studies have examined the defense mechanisms native species employ against non-native predators, particularly in comparison to their effectiveness against native predators (*Davis, Epp & Gabor, 2012*; *Sugiura & Date, 2022*).

The American bullfrog, *Aquarana catesbeiana* (Shaw) (= *Rana catesbeiana* Shaw = *Lithobates catesbeianus* (Shaw)) (Anura: Ranidae), is native to eastern North America (*Ficetola, Thuiller & Miaud, 2007*). *Aquarana catesbeiana* has been intentionally introduced to western North America, South America, Europe, and East–Southeast Asia (*e.g.*, *Ficetola, Thuiller & Miaud, 2007*). *Aquarana catesbeiana* was introduced to Japan from the United States as a food resource in 1918, subsequently escaped from breeding farms, and has since established populations in ponds, lakes, and paddy fields across Japan (*Ota, 2002*; *Matsui & Maeda, 2018*). Female bullfrogs lay eggs in ponds and lakes (*Govindarajulu, Price & Anholt, 2006*), and the larvae feed on tiny invertebrates and algae in water (*Ruibal & Laufer, 2012*). Postmetamorphic juveniles and adults, abundant in and around ponds, can swallow smaller animals (*e.g.*, *Bruneau & Magnin, 1980*; *Flynn, Kreofsky & Sepulveda, 2017*; *Oda et al., 2019*). Previous studies have suggested that the invasion of bullfrogs reduced native terrestrial and aquatic animal populations through direct predation (*Kats & Ferrer, 2003*; *Li et al., 2011*), altered the structure of native aquatic communities (*Gobel, Laufer & Cortizas, 2019*), and affected the behavior of native animals in invaded areas (*Silveira & Guimarães, 2021*). Many studies have investigated the gut or stomach contents of *A. catesbeiana* juveniles and adults, revealing how often they consume native animal species (*e.g.*, *Silva et al., 2009*; *Barrasso et al., 2009*; *Flynn, Kreofsky & Sepulveda, 2017*; *Oda et al., 2019*; *Matsumoto, Suwabe & Karube, 2020*). When a native animal species was found in bullfrog gut or stomach contents, many studies concluded that non-native bullfrogs could have negatively impacted the native animal species (*Hirai, 2005*; *Wu et al., 2005*; *Hirai & Inatani, 2008*; *Silva et al., 2009*; *Barrasso et al., 2009*; *Oda et al., 2019*; *Nakamura & Tominaga, 2021*). However, the finding of a native species in bullfrog gut or stomach contents does not always indicate that bullfrogs frequently prey on that species. For example, high densities of a native animal species can increase encounters with bullfrogs and/or successful predation events by bullfrogs, even when the predation success on the native species by bullfrogs is extremely low. Therefore, it is important to observe defensive responses of native species to bullfrogs to estimate the potential impacts of non-native bullfrogs on native species accurately. However, only a few studies have investigated how native species can defend against non-native bullfrogs (*Sugiura & Date, 2022*).

Adults of whirligig beetles (Coleoptera: Gyrinidae) swim and feed on other arthropods on the water surface of ponds, lakes, and streams (*Sato, 1997a*; *Beutel & Roughley, 2005*; *Yee & Kehl, 2015*). They lay eggs on the submerged parts of plants, their hatched larvae consume other arthropods in water, and the mature larvae emerge onto land to pupate in soil (*Beutel & Roughley, 2005*). Because whirligig beetles are dependent on water, they are potentially vulnerable to aquatic predators such as fish (*van der Eijk, 1986*), newts (*Roşca et al., 2013*), frogs (*Korschgen & Moyle, 1955*; *Stewart & Sandison, 1972*; *McKamie & Heidt, 1974*), birds (*Ikeda, 1956*), and backswimmers (*Härlin et al., 2005*). Whirligig beetles have

evolved various types of anti-predator strategies to evade these predators (*Scrimshaw & Kerfoot, 1987*; *Beutel & Roughley, 2005*; *Dettner, 2019*). For example, the rapid swimming, turning, diving, and aggregation behavior of whirligig beetle adults can reduce their predation risk (*Heinrich & Vogt, 1980*; *Vulinec & Miller, 1989*; *Watt & Chapman, 1998*; *Fish & Nicastro, 2003*; *Romey, Walston & Watt, 2008*). Additionally, adults of the whirligig beetle genera *Gyrinus* Geoffroy and *Dineutes* Macleay emit odoriferous fluids (including norsesquiterpenes; gyrinidal, isogyrinidal, gyrinidon, and gyrinidion) from a pair of pygidial glands (*Meinwald, Opheim & Eisner, 1972*; *Newhart & Mumma, 1978*; *Eisner, Eisner & Siegler, 2005*; *Dettner, 2019*) to deter aquatic predators such as newts (*Benfield, 1972*) and carnivorous fish (*Benfield, 1972*; *Eisner & Aneshansley, 2000*). Since whirligig beetles have been found in gut or stomach contents of the bullfrog *A. catesbeiana* (*Raney & Ingram, 1941*; *Korschgen & Moyle, 1955*; *McKamie & Heidt, 1974*; *Sarashina, 2016*; *Laufer et al., 2021*) and other frogs (*Stewart & Sandison, 1972*), they may be preyed upon by frogs in ponds. Although *Miller, Hendry & Mumma (1975)* suggested that chemicals from adult whirligig beetles are toxic to a frog species *Lithobates pipiens* (Schreber) (= *Rana pipiens* Schreber) (Anura: Ranidae), the effectiveness of chemical defenses against bullfrogs and other frogs has not been quantitatively investigated.

To test whether whirligig beetles can successfully defend against frogs, we examined the effectiveness of defenses of two Japanese whirligig beetle species, *Gyrinus japonicus* Sharp and *Dineutus orientalis* (Modeer), against the black-spotted pond frog *Pelophylax nigromaculatus* (Hallowell) (Anura: Ranidae) and the bullfrog *A. catesbeiana* under laboratory conditions. Both whirligig species, *G. japonicus* (Fig. 1A) and *D. orientalis* (Fig. 1B), are native to East Asia, including Japan, Korea, China, and Russia (*Sato, 1997b*; *Sato, 1997c*; *Nakajima et al., 2020*). Adults of both species swim on the lentic water surface (*Sato, 1997b*; *Sato, 1997c*; *Mitamura et al., 2017*; *Nakajima et al., 2020*) and rest on plant leaves floating on the water surface (Fig. 1; (*Tsuzuki, Taniwaki & Inoda, 2000*). *Pelophylax nigromaculatus* (Fig. 1C), also native to East Asia including Japan, Korea, and China (*Komaki et al., 2015*; *Matsui & Maeda, 2018*), frequently coexists with whirligig beetles *G. japonicus* and *D. orientalis* in ponds in Japan (*Kawahara & Takahashi, 2001*; *Hayashi et al. 2006*; *Hoshizaki Green Foundation, 2006*; *Kawano et al., 2006*). Because juveniles and adults of *P. nigromaculatus* prey on both terrestrial and aquatic insects (*Hirai & Matsui, 1999*; *Sarashina, Yoshihisa & Yoshida, 2011*; *Sano & Shinohara, 2012*; *Sarashina, 2016*), *P. nigromaculatus* potentially attacks *G. japonicus* and *D. orientalis* under field conditions. Furthermore, *A. catesbeiana* (Fig. 1D) has invaded ponds and lakes where native whirligig beetles are found in Japan (*Kurasawa & Okino, 1975*; *Uematsu, 1982*; *Kawahara & Takahashi, 2001*; *Hayashi et al., 2006*; *Hoshizaki Green Foundation, 2006*; *Kawano et al., 2006*; *Endo & Sato, 2010*; *Mitamura & Yoshii, 2011*). In this study, we explored the defensive responses of native whirligig beetles against *P. nigromaculatus* and *A. catesbeiana* and finally discussed the vulnerability of native whirligig beetles to invasive non-native bullfrogs.

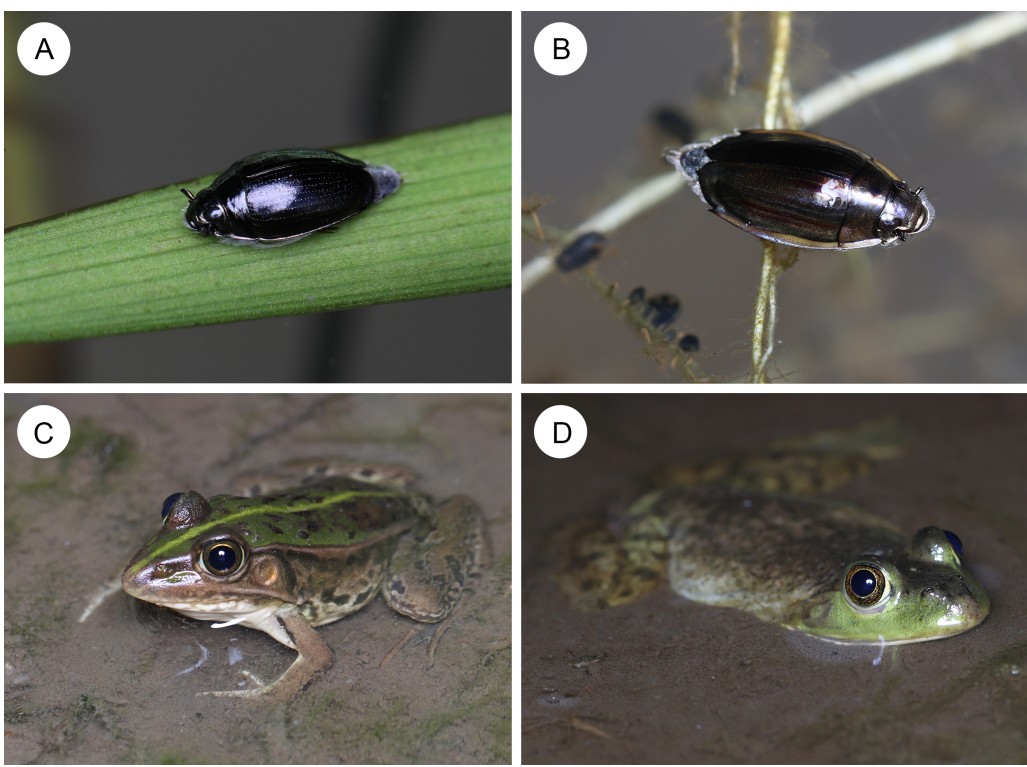

**Figure 1  Whirligig beetles and their potential predators in ponds.** (A) An adult whirligig beetle *Gyrinus japonicus* on a leaf floating on water. (B) An adult whirligig beetle *Dineutus orientalis* on a leaf floating on water. (C) A native pond frog *Pelophylax nigromaculatus*. (D) A non-native bullfrog *Aquarana catesbeiana*. Photo credit: Shinji Sugiura.

## MATERIALS AND METHODS

### Sampling

We collected 30 adult *Gyrinus japonicus* (body weight: 10.0–22.7 mg; body length: 6.0–7.8 mm) and 30 adult *Dineutus orientalis* (body weight: 28.2–62.1 mg; body length: 9.3–11.4 mm) using D-frame nets from ponds in Hyogo, Okayama, and Shimane Prefectures in September–October 2022 and May–September 2023. Adult beetles were housed in plastic cases (length: 250 mm; width: 170 mm; height: 170 mm) with water (depth: 50 mm) under laboratory conditions (25 °C). Newly hatched nymphs of the cockroach *Shelfordella lateralis* (Walker) (Blattodea: Blattidae) were provided as prey. We measured the body weight and length of each beetle to the nearest 0.1 mg and 0.01 mm using an electronic balance (CPA64, Sartorius Japan K.K., Tokyo, Japan) and electronic slide calipers, respectively. We did not use the same individuals in different experiments.

We collected 30 adults and juveniles of *P. nigromaculatus* (body weight: 1,022.5–24,931.8 mg; snout–vent length: 24.6–65.5 mm) from ponds and grasslands in Hyogo, Okayama, and Shimane Prefectures in July–October 2022 and May–September 2023. We also collected 30 juveniles of *A. catesbeiana* (body weight: 3,087.3–89,130.0 mg; snout–vent length: 35.2–101.0 mm) from ponds in Hyogo Prefecture in October 2021, September 2022,
and May–September 2023. Although both juvenile and adult *A. catesbeiana* potentially attack adult whirligig beetles under field conditions, juvenile *A. catesbeiana* were used in this study because of their easy availability. Frogs were housed separately in plastic cages (length: 120 mm; width: 85–190 mm; height: 130 mm) under laboratory conditions (25 °C). Live cockroaches (*i.e.*, nymphs and adults of *S. lateralis*) and mealworms (*i.e.*, larvae of *Tenebrio molitor* Linnaeus (Coleoptera: Tenebrionidae)) were provided as prey. We measured the body weight and snout–vent length of each frog to the nearest 0.1 mg and 0.01 mm using an electronic balance (CPA64; Sartorius Japan K.K., Tokyo, Japan) and electronic slide calipers, respectively. Frogs heavier than 50 g were measured to the nearest 10 mg using an electronic balance (FX–1200i; A&D Company, Limited, Tokyo, Japan). We did not use the same individuals in different experiments. Because *A. catesbeiana* has been designated as an "invasive non-native species" in Japan (*Matsui & Maeda, 2018*), we performed transportation, laboratory keeping, and behavioral experiments of *A. catesbeiana* with permission from the Kinki Regional Environmental Office of the Ministry of the Environment, Government of Japan (Number: 20000085).

## Laboratory observations

Following previous studies (*Sugiura, 2020b*; *Sugiura & Date, 2022*), we observed the behavioral responses of the frogs (*P. nigromaculatus* and *A. catesbeiana*) to adults of whirligig beetles (*G. japonicus* and *D. orientalis*) under laboratory conditions (25 °C) in September–October 2022 and June–September 2023. First, we placed a frog in a plastic cage (length: 120 mm; width: 85–190 mm; height: 130 mm). We did not feed each frog for at least 24 h before our observation to standardize its hunger level (cf. *Honma, Oku & Nishida, 2006*; *Sugiura, 2020b*). Second, we placed a whirligig beetle in the cage with the frog. We recorded the behaviors of the frog and the whirligig beetle using a digital video camera (Handycam HDR-PJ790V; Sony, Tokyo, Japan) and a digital camera (iPhone 12 Pro Max; Apple Inc., Cupertino, CA, USA). We carefully reviewed the footage of recorded behavior under QuickTime Player version 10.5 to analyze how each frog attempted to attack a beetle. If a frog did not attack a beetle within 1 min, we considered the beetle ignored by the frog. When a frog attacked a beetle, we noted whether the frog took the beetle into its mouth. If the frog did take a beetle into its mouth, we then observed whether the frog spat the beetle out. In cases where a frog swallowed a beetle, we considered the beetle successfully eaten by the frog. For any whirligig beetle that was neither attacked nor swallowed, we determined it had been rejected by the frog. Because our focus was on the chemical or physical defenses of whirligig beetles against frogs in this study, we did not investigate whether whirligig beetles could escape frogs on the water surface. For laboratory observations, we used 30 *G. japonicus* (15 tested with *P. nigromaculatus* and 15 with *A. catesbeiana*), 30 *D. orientalis* (15 tested with *P. nigromaculatus* and 15 with *A. catesbeiana*), 30 *P. nigromaculatus* (15 tested with *G. japonicus* and 15 with *D. orientalis*), and 30 *A. catesbeiana* (15 tested with *G. japonicus* and 15 with *D. orientalis*). We determined the sample size based on the minimum number of whirligig beetles collected in the field. Beetles were haphazardly chosen to be provided to *P. nigromaculatus* and *A. catesbeiana*.

All experiments were performed in accordance with the Kobe University Animal Experimentation Regulations (Kobe University's Animal Care and Use Committee, No. 30–01, 2023–03). After completing all observations, we maintained individuals of *P. nigromaculatus* for use in other studies, while individuals of *A. catesbeiana* were euthanized by $CO_2$ asphyxiation.

## Data analysis

We used $t$-test to compare the body size (body weight and length) of whirligig beetles that were provided to *P. nigromaculatus* and *A. catesbeiana*. Welch's $t$-test was used due to the detection of unequal variances between the groups (cf. *Sokal & Rohlf, 2001*). We also used Fisher's exact test to compare the rejection rates between *P. nigromaculatus* and *A. catesbeiana* for each whirligig beetle species. Furthermore, we used generalized linear models (GLMs) with a binomial error distribution and logit link function to determine the effects of body size on the probability of rejection of whirligig beetles by frogs (cf. *Sugiura, 2018*). The rejection (1) or predation (0) of a whirligig beetle (*G. japonicus* or *D. orientalis*) by a frog was used as the response variable. Beetle weight, frog weight, and the beetle weight × frog weight interaction were used as explanatory variables. All analyses were performed at the 0.05 significance level using R version 4.3.0 (*R Core Team, 2023*).

## RESULTS

Almost all individuals of *P. nigromaculatus* attacked adults of *G. japonicus* (Table 1; Fig. 2). Nine frogs (60%) successfully swallowed adults of *G. japonicus* (Table 1; Fig. 2). However, five frogs (36%) rejected adults of *G. japonicus* after attacking them (Table 1; Fig. 2); two frogs stopped attacking the beetles immediately after their tongues contacted them, while three frogs spat out the beetles within 2 s of taking them into their mouths (Figs. 2, 3A). The body size of *G. japonicus* and *P. nigromaculatus* did not significantly influence the rejection rate of *G. japonicus* by *P. nigromaculatus* (Table 2).

All individuals of *P. nigromaculatus* attacked adults of *D. orientalis* (Table 1; Fig. 2). Nine frogs (60%) successfully swallowed adults of *D. orientalis* (Table 1; Fig. 2). However, six frogs (40%) rejected adults of *D. orientalis* after attacking them (Table 1); one frog stopped attacking the beetle immediately after its tongue contacted it, while five frogs spat out the beetles within 2 s of taking them into their mouths (Figs. 2, 3B). The body size of *D. orientalis* and *P. nigromaculatus* did not significantly influence the rejection rate of *D. orientalis* by *P. nigromaculatus* (Table 2).

All individuals of *A. catesbeiana* attacked and swallowed adults of *G. japonicus* and *D. orientalis* (Table 1; Fig. 2). No bullfrogs rejected the beetles after attacking them. The GLM analyses were not conducted on *A. catesbeiana*, as no individuals rejected the beetles.

The body weights and lengths of *G. japonicus* provided to *P. nigromaculatus* did not significantly differ from those provided to *A. catesbeiana* ($t$-test, $P = 0.85-1.00$). The body weights of *D. orientalis* (mean ± standard errors, $38.5 \pm 2.2$ mg) provided to *P. nigromaculatus* significantly differed from those of *D. orientalis* ($45.1 \pm 1.9$ mg) provided to *A. catesbeiana* ($t$-test, $P = 0.03$), although the body lengths of *D. orientalis* given to
**Table 1** Behavioral responses of frogs to whirligig beetles.

| Frog species | Beetle species | Attack | | | Ignore | Total |
|---|---|---|---|---|---|---|
| | | Swallow | Spit out | Stop attack | | |
| *Pelophylax nigromaculatus* | *Gyrinus japonicus* | 9 | 3 | 2 | 1 | 15 |
| | *Dineutus orientalis* | 9 | 5 | 1 | 0 | 15 |
| *Aquarana catesbeiana* | *Gyrinus japonicus* | 15 | 0 | 0 | 0 | 15 |
| | *Dineutus orientalis* | 15 | 0 | 0 | 0 | 15 |

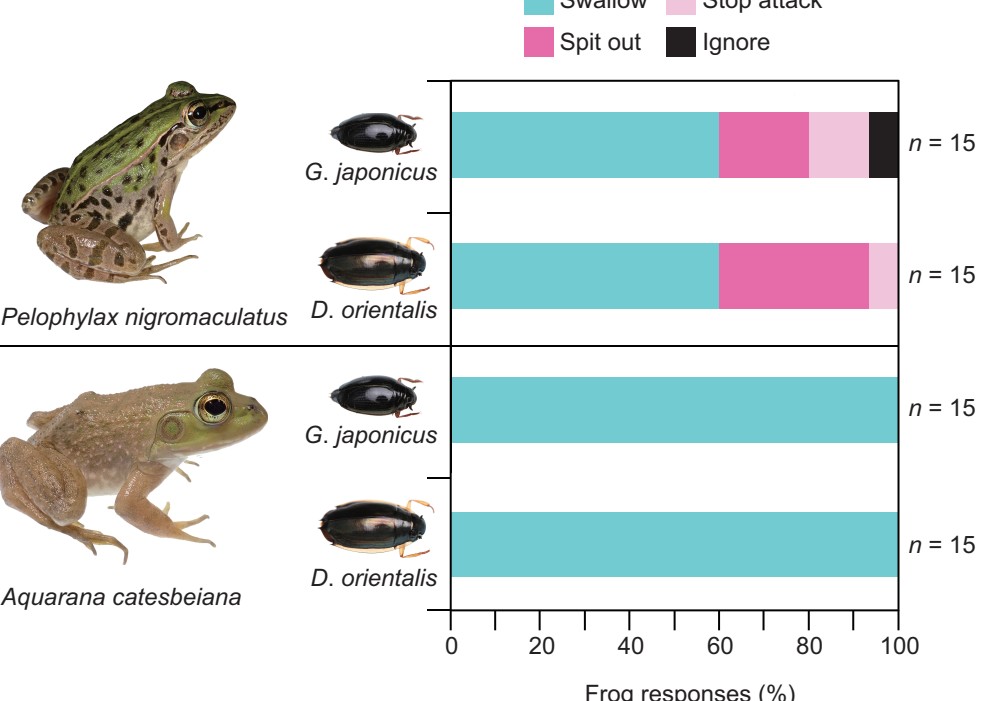

**Figure 2** **Behavioral responses of the native pond frog *Pelophylax nigromaculatus* and the non-native bullfrog *Aquarana catesbeiana* to native whirligig beetles *Gyrinus japonicus* and *Dineutus orientalis*.** "Swallow": frogs successfully swallowed beetles. "Spit out": frogs spat out beetles immediately after they took them into their mouths. "Stop attack": frogs stopped attacking beetles immediately after their tongues contacted them. "Ignore": frogs did not attack beetles. Photo credit: Shinji Sugiura.

*P. nigromaculatus* did not significantly differ from those given to *A. catesbeiana* ($t$-test, $P = 0.14$).

The rejection rates of whirligig beetles significantly differed between the two frog species, *P. nigromaculatus* and *A. catesbeiana* (Fisher's exact test: *G. japonicus*, $P = 0.0169$; *D. orientalis*, $P = 0.0169$).

# DISCUSSION

Some native species can survive the predation pressures of non-native species by using pre-existing anti-predator strategies (*Davis, Epp & Gabor, 2012*; *Carthey & Banks, 2014*;

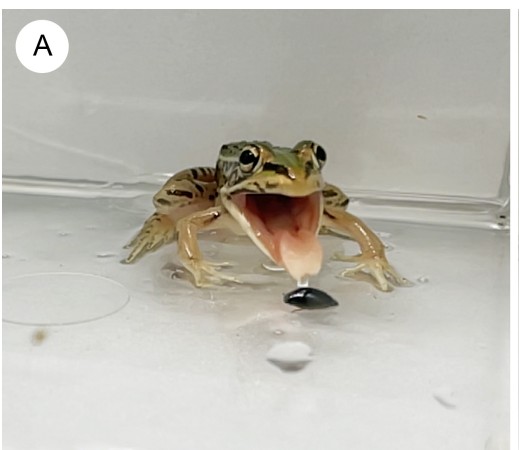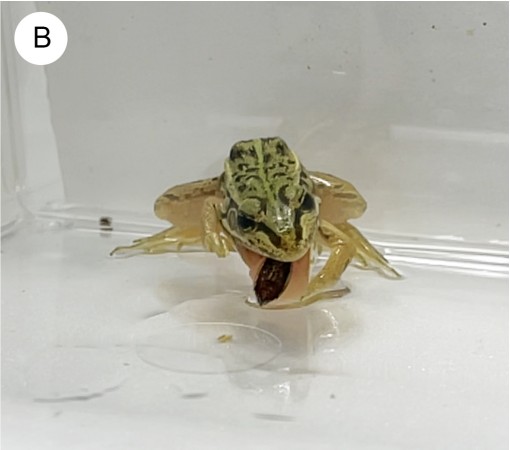

**Figure 3** **The native pond frog *Pelophylax nigromaculatus* rejecting native whirligig beetles.** (A) A frog rejecting an adult *Gyrinus japonicus*. (B) A frog rejecting an adult *Dineutus orientalis*. Each frog spat out the whirligig beetle immediately after taking it into the mouth. Photo credit: Shinji Sugiura.

**Table 2** **Results of generalized linear models (GLMs) identifying factors affecting whether the native pond frog *Pelophylax nigromaculatus* rejected whirligig beetles.**

| Beetle species | Response variable | Explanatory variable | Coefficient estimate | SE | z value | P value |
|---|---|---|---|---|---|---|
| *Gyrinus japonicus* | Rejection | Intercept | −24.51 | 19.92 | −1.23 | 0.219 |
| | | Beetle size (weight) | 1.598 | 1.263 | 1.265 | 0.206 |
| | | Frog size (weight) | 0.005521 | 0.005701 | 0.969 | 0.333 |
| | | Beetle size × frog size | −0.0003592 | 0.0003535 | −1.016 | 0.310 |
| *Dineutus orientalis* | Rejection | Intercept | 22.06 | 21.50 | 1.026 | 0.305 |
| | | Beetle size (weight) | −0.4700 | 0.5703 | −0.824 | 0.410 |
| | | Frog size (weight) | −0.003471 | 0.004905 | −0.708 | 0.479 |
| | | Beetle size × frog size | 0.00006821 | 0.0001344 | 0.507 | 0.612 |

*Sugiura & Date, 2022*). To test how native whirligig beetles defend against native and non-native predators, we investigated the defensive effectiveness of native *G. japonicus* and *D. orientalis* against native *P. nigromaculatus* and non-native *A. catesbeiana* under laboratory conditions. While some adults of *G. japonicus* and *D. orientalis* could successfully defend against *P. nigromaculatus*, none could evade predation by *A. catesbeiana*. The chemical defenses of *G. japonicus* and *D. orientalis* provided some protection against individuals of *P. nigromaculatus* but were entirely ineffective against *A. catesbeiana*. Consequently, pre-existing anti-predator strategies of *G. japonicus* and *D. orientalis* failed to repel *A. catesbeiana*.

Whirligig beetles can be attacked and eaten by various types of aquatic predators such as fish (*van der Eijk, 1986*), amphibians (*Korschgen & Moyle, 1955*; *Stewart & Sandison, 1972*; *McKamie & Heidt, 1974*), birds (*Ikeda, 1956*), and arthropods (*Härlin et al., 2005*). Whirligig beetles have evolved various defense strategies to deter these aquatic predators (*Beutel & Roughley, 2005*). Rapid swimming, turning, diving, and aggregation behaviors

of whirligig beetle adults (*Heinrich & Vogt, 1980*; *Vulinec & Miller, 1989*; *Fish & Nicastro, 2003*) function as "primary defenses" before predators physically contact them (definition in *Ruxton et al., 2018*). Chemicals emitted by whirligig beetle adults (*Benfield, 1972*; *Eisner & Aneshansley, 2000*) act as "secondary defenses" once predators physically contact them (definition in *Ruxton et al., 2018*). In this study, we focused on the secondary defenses of adult whirligig beetles but did not investigate the escape behavior of whirligig beetles on the water surface. Because the primary defenses of adult whirligig beetles may play important roles in escaping from frogs, the defense success rates in our observations may be underestimated.

Previous studies have shown that chemicals prevent whirligig beetle adults from being preyed upon by fish and newts (*Benfield, 1972*; *Eisner & Aneshansley, 2000*). In this study, we showed that some individuals of *P. nigromaculatus* rejected whirligig beetle species *G. japonicus* or *D. orientalis*. Some frogs stopped attacking whirligig beetles immediately after their tongues contacted them, while others spat them out within 2 s of taking them into their mouths. These behaviors have frequently been observed in *P. nigromaculatus* as a rejection of chemically defended insects (*Sugiura, 2018*; *Sugiura & Hayashi, 2023*). Therefore, norsesquiterpenes (*e.g.*, gyrinidal, isogyrinidal, gyrinidon, and gyrinidion) emitted by adults of *G. japonicus* and *D. orientalis* could play an important role in deterring *P. nigromaculatus*. Although previous studies indicated that prey or frog size could influence the success of prey chemical defenses against frogs (*Sugiura & Sato, 2018*; *Sugiura, 2018*; *Sugiura & Tsujii, 2022*), we could not detect any effects of whirligig beetle and/or frog size on the defensive success against *P. nigromaculatus* in this study. Other factors, such as variation in beetle chemicals, may influence the defensive success of whirligig beetles. In addition, no adults of whirligig beetles have been found in the gut or stomach contents of field-collected *P. nigromaculatus* (*Hirai & Matsui, 1999*; *Sarashina, Yoshihisa & Yoshida, 2011*; *Sano & Shinohara, 2012*; *Sarashina, 2016*), suggesting that defensive behaviors (*e.g.*, rapid swimming, turning, and diving) as well as chemicals of adult *G. japonicus* and *D. orientalis* may play important roles in escaping from *P. nigromaculatus* in ponds and lakes.

Whirligig beetles were found in the gut or stomach contents of *A. catesbeiana* in both bullfrogs' native habitats (*Raney & Ingram, 1941*; *Korschgen & Moyle, 1955*; *McKamie & Heidt, 1974*) and invaded areas (*Sarashina, 2016*; *Laufer et al., 2021*). In Japan, the native whirligig beetle species *D. orientalis* was found in the gut content of a field-collected *A. catesbeiana* (*Sarashina, 2016*), suggesting that *A. catesbeiana* preyed on *D. orientalis* under field conditions. Our laboratory observations showed that all adults of *G. japonicus* and *D. orientalis* were swallowed by juvenile *A. catesbeiana* (Table 1; Fig. 2). Adult bullfrogs, with a snout–vent length (SVL) of 111–183 mm (*Matsui & Maeda, 2018*), were much larger than the juveniles used in this study (SVL 35.2–101.0 mm), suggesting that bullfrogs of any size can easily eat whirligig beetles. The chemical and other secondary defenses of *G. japonicus* and *D. orientalis* are not able to deter *A. catesbeiana*. Because other species of the genera *Gyrinus* and *Dineutus* are found in *A. catesbeiana*'s native distribution range (*Roberts, 1985*; *Oygur & Wolfe, 1991*), *A. catesbeiana* has an evolutionary history with beetles of the genera *Gyrinus* and *Dineutus* that enables it to counter their chemical defenses. However, the

East Asian species *G. japonicus* and *D. orientalis* may not share sufficient history with *A. catesbeiana* to strengthen their defenses against it.

## CONCLUSIONS

Our results indicate that native whirligig beetles are vulnerable to predation by non-native bullfrogs in local ponds and lakes across Japan. Despite this, historical evidence suggests that adult *G. japonicus* and *D. orientalis* have temporally coexisted with *A. catesbeiana* in the same habitats within Japan (*Kurasawa & Okino, 1975*; *Uematsu, 1982*; *Kawahara & Takahashi, 2001*; *Hayashi et al., 2006*; *Hoshizaki Green Foundation, 2006*; *Kawano et al., 2006*; *Endo & Sato, 2010*; *Mitamura & Yoshii, 2011*). Further research is needed to explore whether native whirligig beetles can persist under the predation pressures of *A. catesbeiana* under field conditions.

## ACKNOWLEDGEMENTS

We thank K. Okai and T. Tokuhira for providing us with information on the sampling of whirligig beetles. We also thank T. Daijima and H. Uchida for their help in maintaining the frogs. We utilized an artificial intelligence chatbot, GPT-4 (https://chat.openai.com), for manuscript correction.

### Funding

This study was supported by a Grant-in-Aid for Scientific Research (JSPS KAKENHI Grant number JP19K06073) and funds from the Hoshizaki Green Foundation. The funders had no role in study design, data collection and analysis, decision to publish, or preparation of the manuscript.

### Grant Disclosures

The following grant information was disclosed by the authors:
Scientific Research (JSPS KAKENHI): JP19K06073.
The Hoshizaki Green Foundation.

### Competing Interests

The authors declare there are no competing interests.

### Author Contributions

- Shinji Sugiura conceived and designed the experiments, performed the experiments, analyzed the data, prepared figures and/or tables, authored or reviewed drafts of the article, and approved the final draft.
- Masakazu Hayashi conceived and designed the experiments, prepared figures and/or tables, authored or reviewed drafts of the article, and approved the final draft.

## Animal Ethics

The following information was supplied relating to ethical approvals (i.e., approving body and any reference numbers):

The authors performed the experiments in accordance with the Kobe University Animal Experimentation Regulations (Kobe University's Animal Care and Use Committee, No. 30–01, 2023–03).

## Field Study Permissions

The following information was supplied relating to field study approvals (i.e., approving body and any reference numbers):

Kinki Regional Environmental Office of the Ministry of the Environment, Government of Japan (Number: 20000085).

## Data Availability

The raw data are available at figshare: Sugiura, Shinji; Hayashi, Masakazu (2024). Data from: Defenses of whirligig beetles against native and invasive frogs. figshare. Dataset. https://doi.org/10.6084/m9.figshare.24746151.

## Supplemental Information

Supplemental information for this article can be found online at http://dx.doi.org/10.7717/peerj.17214#supplemental-information.

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
