# Peer review of "Defenses of whirligig beetles against native and invasive frogs"

_PeerJ, doi:10.7717/peerj.17214_

## Round 0.1 · original submission · Minor Revisions

The reviewers agree that your manuscript is interesting, well-prepared, and adds valuable information to the field. They all spot some formatting and citation issues, please address them all.

Cheers,

Reviewer 1 ·

Basic reporting

The manuscript was well written and easy to follow. This is the first time I have seen a manuscript that was checked by AI for English. Interesting.

Line 132. replace "adults of" with "adult"
Line 133. report on what body of water specimens were collected and how (D-net?)
Line 66-71. too many references (26!) for a fairly minor point. Perhaps report just the 5 most important or recent?
Line 124. Get rid of last citation that is not published (Mari).

Experimental design

Overall the design is good. Even though it is explained why the whirligigs are presented out of water, it does reduce the importance of this study. Perhaps the whirligigs are never eaten by frogs in the wild because of their effective grouping behavior. Sample sizes seem fine. Methods are described well. Although I'm not sure why they used juvenile bullfrogs (L.c.) rather than adult.

Validity of the findings

Table 1 presentation is confusing and perhaps not even necessary. Perhaps should be arranged with first two lines as beetle species and second two lines as frog species only rather than a 2 x 2 table? Or perhaps eliminate this table and replace it with a few sentences in the results section saying that there were no significant differences in sizes of whirligigs. Of course the species of frogs are going to be different sizes, so no need to report comparative stats on this, just the range.

Otherwise, validity is good and conclusions are well stated.

·

Basic reporting

I would like to express my gratitude to the Editor for entrusting me with the review of the manuscript. It has been a rewarding experience to evaluate the submission, which I find to be both intellectually stimulating and scientifically sound. The manuscript excels in clarity and professionalism, maintaining an adequate standard of English throughout. This document stands out as self-contained, with results directly addressing the stated hypotheses. The professional structure of the article, including well-constructed figures and tables, contributes to its overall coherence. The author has provided ample literature references (in some cases in excess; see comments below), establishing a solid background in the field. The scientific relevance and quality of the work are evident, underscoring its significance in the field of invasion ecology. It is an engaging and valuable contribution to the scientific literature on biological invasions.

Experimental design

The manuscript presents a well-conceived and straightforward experimental design that tackles fundamental questions in invasion biology. Drawing from prior data, including contributions from the authors, the study comprehensively explores the advantages of non-native organisms over their native counterparts. Central to the investigation is a key question regarding the success of non-native organisms, with a novel twist: the study delves into the realm of chemical defenses. This innovative approach enhances the significance of the research, shedding light on the mechanisms driving the success of non-native species. Notably, the study employs the American bullfrog, a widely invasive and globally important species, as the model organism. This choice adds depth to the research, as the bullfrog's expansive presence enhances the broader implications of the findings.
Despite its simplicity, the experiment yields compelling results, and the methodology is sufficiently detailed for replication. The study maintains high technical and ethical standards, making it a valuable contribution to invasion biology. By combining a thoughtful experimental design with a unique focus on chemical defenses and utilizing the globally significant bullfrog as the model organism, this research provides meaningful insights into the dynamics of biological invasions.

Validity of the findings

It is crucial to assess differences that impact the interactions between non-native organisms and native species in invaded areas. These evaluations are fundamental for understanding the competitive advantages that invasive organisms may gain, providing valuable insights for the management and conservation of native species.

The need for an in-depth understanding of the bullfrog becomes particularly significant, given its status as a globally significant invasive species. A comprehensive knowledge of the bullfrog's natural history is essential for developing effective control plans in various affected regions. A profound understanding of its biology and behavior serves as a necessary foundation for implementing management strategies that are both effective and ecologically balanced.

Additional comments

Given the manuscript's strong scientific relevance, overall quality, and well-crafted writing, I am inclined to recommend its acceptance with minor revisions.I have outlined minor comments to the authors below, to improve the manuscript quality.
Line 15: I recommend aligning with the recent trend of avoiding the use of potentially confusing terms for non-native species, such as 'alien' and 'exotic.' I recommend thoroughly reviewing and adjusting the terminology throughout the entire manuscript to align with the guidelines presented in Soto et al.'s preprint titled 'Taming the terminological tempest in invasion science' (DOI: 10.32942/X24C79). This adjustment will not impact the content but will enhance the overall quality of the manuscript.
Line 16: The species name Lithobates catesbeianus has been changed to Aquarana catesbeiana. While this change occurred some time ago, current articles are rapidly adopting the updated nomenclature. This can be verified on the Frost 2024 website (https://amphibiansoftheworld.amnh.org). Nevertheless, I appreciate the authors including synonymy in the species presentation or keywords.
Lines 29-31: I recommend a revision to better discuss the implications of your results. I find the conclusion of the abstract somewhat understated given the study's significance.
Line 39: I suggest changing the term 'devastated,' as it appears overly catastrophic. The concept seems appropriate; I would recommend using a different word to convey it.
Line 39: "predators ........ through direct predation" The quality of this sentence could be enhanced. Instead of repeating 'predation,' consider using 'consumption' to convey the idea of predators affecting native fauna directly.
Lines 49-51: Without diminishing the relevance of your manuscript, I note a potential error in how you articulate what you are testing in your experiment. You did not specifically test how non-native predators can negatively impact native species; rather, you investigated a potential mechanism. The broader implications of your findings in relation to such impacts are more fitting for the discussion section.
An important recommendation for this section and throughout the manuscript is to avoid over-reliance on self-citations.
Line 52: “=” NOT “formerly called” (for clearly explaining the synonymy)
Lines 53-54: In this sentence, the citations seem inappropriate. I suggest deleting “Lowe et al. 2000”. After the species name, you may include the author's name or leave it blank. However, citing the book on invasives seems inappropriate.
Lines 56-58: There seems to be an overuse of citations here. I suggest using only the first one, as it encompasses all the continents mentioned.
Lines 58-60: These two sentences can be condensed into one for brevity. I recommend omitting the use of 'for example.
Lines 61-64: Remove 'than themselves' and try to reduce the number of citations to the essential ones. Another important consideration is that bullfrog tadpoles are also predators or, at the very least, omnivores.
Quiroga, L., Olivencia, N., Ray, M., Wetten, P., Rodriguez, Y., Traverso, J. A., ... & Sanabria, E. (2022). Diet of Bullfrog Tadpoles Lithobates catesbeianus, Shaw 1802, an Invasive Species from Monte Desert. Journal of Herpetology, 56(3), 312-317.
Ruibal, M., & Laufer, G. (2012). Bullfrog Lithobates catesbeianus (Amphibia: Ranidae) tadpole diet: description and analysis for three invasive populations in Uruguay. Amphibia-Reptilia, 33(3-4), 355-363.
Line 65: Delete 'might have.' The reported effects are significant enough not to be understated.
Line 67: Adriaens et al. 2013 provides an interesting risk analysis, but I understand it may not contribute new information to support your statement. Therefore, I suggest that this citation is not suitable here.
Lines 68-77: I appreciate the inclusion of field evidence by the authors to justify their experiment. However, I note an abundance of citations related to diet. I recommend reviewing them, as papers like Jancowski & Orchard, 2013, and Laufer et al., 2021, include reviews of other studies that are also cited. Perhaps, by consolidating based on this criterion, you can reduce the number of citations, enhancing the overall quality of the manuscript.
Lines 77-85: Regarding your statement, 'When a native animal species was found in bullfrog gut or stomach contents...,' I would like to emphasize that relying solely on dietary studies might be limiting. There are valuable studies on changes in community structures, trophic network structures, behaviors, and developmental impacts on native species, among others, after bullfrog invasion.
Line 114: I suggest that the authors specify which species of frog they are referring to.
Lines 133-134: I have some questions here that would benefit from clarification. How many years have passed since this invasion? How long have they coexisted? Could there have been local microevolutionary changes contributing to tolerance over time?
This aspect could also be addressed in the discussion section.
Line 138: Evaluated (Explored) NOT “observed”
MATERIALS AND METHODS
Overall, for this section, I recommend providing more detailed descriptions of the experimental procedures, including times, dimensions, etc., as well as elaborating on the methods used to obtain the response variables.
Lines 151-152: Adding the brand and model of the scale and caliper may not contribute significantly to the content
Lines 154-156: It would be beneficial for the authors to explain why they chose to use juveniles and not other ontogenetic stages for conducting the experiments.
Lines 189-194: Here, the authors are repeating information already stated above. It would be advisable to avoid redundancy.
Line 200: I believe this is an example of unnecessary over-reliance on self-citations.
Lines 203-204: I suggest the authors relocate this sentence to the end of the data analysis section.
Lines 205-208: I assume you compared mean differences between groups. Did you have different variances? Please explain why you chose this test and provide relevant bibliographic citations.
Line 209 and throughout the manuscript: I believe it is not always necessary to explicitly mention that it is a frog and whether it is native or not. For example, here you could eliminate “the native frog”.
Line 212: the effects of body size on the probability of rejection NOT the effects of body size on the rejection
Lines 217-218: I would suggest moving this to the Results section.
RESULTS
60% seems to be better than 60.0% (the same for other mentioned percentages)
Lines 239-241: These results do not correspond to something explained in the methods section, I recommend providing the necessary details or context in that section for clarity.
DISCUSSION
Lines 244-245: While I understand the authors may not have confusion, the mix of individual and population costs is not clearly presented and could be confusing for the reader. I suggest rewriting this section for clarity
Line 249: It would be beneficial to discuss and present possible explanations for why this occurs for 'some' but not all.
Line 250: Avoid citing figures or tables in the discussion.
Lines 252-253: Does this happen despite the prey's long period of coexistence with bullfrogs? It would be beneficial to define this aspect in the system of origin of the specimens used.
Line 268: This is not a limitation of the study. It's good to mention it but not categorize it as a limitation.
Lines 282-283: Can you discuss why you think this happened? Could it be a limited range of individuals that were selected?
Lines 301-302: I disagree with this conclusion. Predation doesn't always have a negative effect on populations. This would be a different level of analysis that goes beyond the scope of your study.
Lines 307-308: Could this phenomenon occur with all local native species, or is it something specific to aquatic insects?
Lines 315-316: I understand that this may not be accurate, as there is a substantial amount of data on this topic. Numerous notes can be found in HerpReview, for example.
Lines 319-322: I believe the conclusion of the manuscript could be enhanced by providing a clearer explanation of the significance of the observations.

Reviewer 3 ·

Basic reporting

The article addresses all the key points effectively. It is coherent and well-written, conveying a very clear message with an appropriate structure. The hypothesis presented is clear and well-constructed, and it undergoes thorough analysis. While the context and references are presented adequately, I have observed a significant number of bibliographic citations in the introduction. I would suggest reducing the number of citations and focusing on maintaining the main references.

Experimental design

The research question is clear, concrete and effectively addressed through the proposed methodology. The methodology is appropriate; however, I recommend providing more detailed explanations for certain aspects of the laboratory procedures. Specifically, I suggest details which are the recorded behaviors that are then analyzed and presented in the results.

Validity of the findings

The statistical analyzes are well presented, and the discussion and conclusions logically stem from the results acquired, aligning seamlessly with the posed question.

Additional comments

The article titled 'Defenses of whirligig beetles against native and aliens frogs' analyzes differences in the defensive responses of two species of Gyrinidae to two species of anurans, one native and the other exotic (bullfrog). The article is concrete and well presented, providing evidence that allows us to evaluate the effects and mechanisms of one of the most dangerous invasive species.

Annotated reviews are not available for download in order to protect the identity of reviewers who chose to remain anonymous.

---

## Round 0.2 · accepted · Accept

I apologize for the delay, one of the reviewers took more time to respond than usual. You can find their comments below, they are happy with the modifications you have made to the manuscript and think it is suitable for publication. I also enjoyed reading your work and appreciate the approach.

Reviewer 1 ·

Basic reporting

I have read over the revised manuscript and am satisfied with the modifications made by the authors to address the various reviewer concerns. The paper now looks to be in good shape and is ready for publication in my opinion in regards to the three criteria (basic reporting, experimental design, and validity of findings).

The only minor thing that might be nice to see them address (in a sentence) would be the age difference in the two types of frogs. How do they think that might influence their findings? I could see that an older frog could have learned to avoid noxious whirligig chemicals in the field more than a naive young frog. Several studies of predators on whirligigs show that there is a strong learning effect on predators: they will learn to avoid whirligigs after sampling one or two. I don't need to see the manuscript again, but if the editor agrees, this might be an important point to make in the Discussion.

Experimental design

no comment -- see above

Validity of the findings

no comment -- see above.

Reviewer 3 ·

Basic reporting

no comment

Experimental design

no comment

Validity of the findings

no comment

Additional comments

The authors have corrected all the aspects indicated in the first revision. This is a well-written and substantiated article, with novel results that contribute to understanding the impacts and mechanisms underlying the bullfrog invasion.